# Stability of a Melt Pool during 3D-Printing of an Unsupported Steel Component and Its Influence on Roughness

**DOI:** 10.3390/ma13030808

**Published:** 2020-02-10

**Authors:** Mateusz Skalon, Benjamin Meier, Andreas Gruberbauer, Sergio de Traglia Amancio-Filho, Christof Sommitsch

**Affiliations:** 1IMAT Institute of Materials Science, Joining and Forming, Graz University of Technology, Kopernikusgasse 24/1, 8010 Graz, Austriasergio.amancio@tugraz.at (S.d.T.A.-F.);; 2Joanneum Research, Materials, Institute for Laser and Plasma Technology, Leobner Straße 94, 8712 Niklasdorf, Austria

**Keywords:** laser powder bed fusion, downskin, melt pool, stability, printing angle, roughness, steel

## Abstract

The following work presents the results of an investigation of the cause–effect relationship between the stability of a melt pool and the roughness of an inclined, unsupported steel surface that was 3D-printed using the laser powder bed fusion (PBF-L/M) process. In order to observe the balling effect and decrease in surface quality, the samples were printed with no supporting structures placed on the downskin. The stability of the melt pool was investigated as a function of both the inclination angle and along the length of the melt pool. Single-track cross-sections were described by shape parameters and were compared and used to calculate the forces acting on the melt pool as the downskin was printed. The single-melt track tests were printed to produce a series of samples with increasing inclination angles with respect to the baseplate. The increasing angles enabled us to physically simulate specific solidification conditions during the sample printing process. As the inclination angle of the unsupported surface increased, the melt-pool altered in terms of its size, geometry, contact angles, and maximum length of stability. The balling phenomenon was observed, quantified, and compared using roughness tests; it was influenced by the melt track stability according to its geometry. The research results show that a higher linear energy input may decrease the roughness of unsupported surfaces with low inclination angles, while a lower linear energy input may be more effective with higher inclination angles.

## 1. Introduction

Laser powder bed fusion methods (PBF-L/M) are becoming more and more important tools in modern manufacturing chains due to the unique manufacturing capabilities they offer. Among these, the most important is the nearly unlimited freedom of design they offer, which allows users to manufacture parts with optimized topologies, depending on the working conditions. This freedom, however, is hindered by the fact that the unsupported overhanging planes exhibit gradually worsening surface quality [1]. In the state of the art, when a plane is printed at a low inclination angle (α) with respect to the printing plane (X, Y dimensions), the supporting structures usually have to be printed to serve as both a solidification base and heat sink [2,3,4]. In most cases, this is necessary for angles lower than 30–45° (where 90° refers to a surface perpendicular to the X–Y printing plane). This limit originates from the low powder thermal conductivity with respect to the bulk counterpart, which is an inherent material property. As a result, the heat originating from the laser beam cannot easily be dissipated [4,5,6,7,8,9,10]. When the temperature of the powder bed exceeds the alloy melting point, T_m_, the powder particles melt and adhere to each other. Therefore, the effective thermal conductivity is again set to that of the dense material [2]. 

Powder melting leads to shrinkage, resulting in a change in the geometry of the structure. According to Wang et al. [11], the inhomogeneous heat outflow results in a bimetallic effect (warpage) of a layer printed at a high α. The higher the inclination angle, the stronger the warping will be, and the more irregular and rougher the downskin will become. In addition, Panwisawas et al. [12] used computational fluid dynamics to show that the pattern flow of a melt pool is governed by various forces, such as the vapor pressure, gravitational force, and capillary and thermal capillary forces exerted on the metallic/gaseous interface. The molten powder forms a pool of molten metal with the surrounding dynamic melt flow and dynamic flow of gases/vapors during the printing process [13,14,15,16]. A moving, focused laser beam leaves a molten track behind; the track should be considered as an elongated, liquid stream extending across the supporting surface in the shape of a cylinder [7,14,17]. Such an elongated cylinder (melt pool) is highly unstable due to the Plateau–Rayleigh instability [7,18]. Gusarov and Smurov [7] identified the segmental cylinder stability of a liquid on a solid substrate and used this value to approximate melt-pool behavior. Moreover, several works on the evaluation and modeling of the melt-track formation have recently been published [14,15,16,19,20]. In all these works, a thin layer of a powder was melted on flat support plates to observe the melt-track behavior. However, none of these experiments enabled the authors to fully and physically describe the melt-pool stability, in part because the melt-pool had a limited connection to the base plate in the case of these inclined printing planes. In addition, no cause–effect relationship could be described between the cylinder stability and the downskin roughness. In this study, singular tracks were printed under conditions resembling true production conditions, and single-track cross-sections (STCS) were described with shape parameters to identify the causes of downskin roughening. In this study, the influence of the inclination angle on the stability of the melt-pool is described, as well as the effects of this stability on the roughness of unsupported surfaces printed using the PBF-L/M process. 

## 2. Experimental Approach

### Base Material

Gas-atomized AISI 316L austenitic stainless-steel powder was selected for the purposes of this study (Figure 1). The chemical composition of the powder is presented in Table 1. The grain size distribution is presented in Figure 1a.

Cuboidal samples (10 × 10 × 5 mm^3^) were printed with stepwise-increasing inclination angles of α = 90°, 80°, 70°, 60°, 50°, 45°, and 40°. No support was provided beneath the bottom surface, allowing the downskin to evolve freely (Figure 2a). All the samples were manufactured using a Farsoon FS121M machine (Farsoon Europe, Stuttgart, Germany). The utilized printing parameters are listed in Table 2. The production parameters were selected based on the parameters suggested by the powder manufacturer (series A) and modified to promote the creation of an elongated melt pool (series B). The parameters in series B were calculated to achieve the same linear energy value (E_1_ = laser power/laser speed).

The series A parameters were changed in such a way as to hold the linear energy constant and to alter the melt-pool length only by varying both the laser power and the spot speed. Both series of samples (A and B) were printed using separate printing processes. Therefore, the compared samples were placed in the same positions on the build-platform. Roughness tests were performed using a Hommel-Etamic Waveline W20 (Jenoptik AG, Jena, Germany) according to DIN EN ISO 4287, DIN EN ISO 13565, and DIN EN 10049 standards. Microscopic analyses were performed using a light optical microscope (LOM, Zeiss, Oberkochen, Germany) and a TESCAN MIRA3 field-emission scanning electron microscope (SEM, TESCAN Brno s.r.o., Brno, Czech Republic). A physical simulation of the melt-pool behavior while creating samples with increasing inclination angles was carried out using the support plate, as shown in Figure 2b. A series of single lines were printed using the following parameters: Series A: 70 W, 990 mm·s^−1^; and series B: 85 W, 1200 mm·s^−1^, with a focused laser beam applied to create a border contour. The focus diameter of the laser beam in both cases was 80 μm. Six samples according to Figure 2b were printed using each set of parameters. Each line was printed on a surface with an incrementally increasing slope angle (β = 0°, 10°, 20°, 30°, 40°, 50°, 60°, and 70°), as presented in Figure 2. Taking this approach allowed us to print single tracks on top of the inclined surfaces of samples that were printed in the same process. 

Next, the samples were cut to obtain cross-sections of the melt tracks for microscopic characterization. Two photos of cross-sections were made and investigated for each line. Taking such an approach allowed us to obtain a population of 12 single-track cross-sections (STCS) for each line and show how the singular track behaves while printing the inclined, unsupported surface.

As presented in Figure 3a, the process of printing the single track on an inclined plane can influence its shape parameters. In order to investigate these, the traditional approach of printing single tracks on a horizontal plane was modified, and a schematic of this modified approach is presented in Figure 3b. In the given case, the liquid cylinder was placed on an incline and, therefore, additional forces were considered (Figure 3b). A variety of forces act on a melt pool during the PBF-L/M process, including laser beam pressure, Maragoni convection, gas flow, and pressure changes [14,15,16]. For simplification in this study, the melt pool was considered as a liquid semi-cylinder attached to a flat, inclined surface that was influenced only by gravitational and surface tension forces.

The singular tracks were cut, and their cross-sections were described by the following numerical parameters: Area “A;” height “h,” which was understood as the shortest distance between the X–Y-line and Z-line; melt-pool base width “L;” radius “R,” which was understood as the radius of a circle described by *x*-, *y*-, and *z*-coordinates; Θ_rec_—receding wetting angle; Θ_adv_—advancing wetting angle; and Φ —the specific angle defining the stability of a cylinder, which is perpendicular to the contact line (Figure 3). 

## 3. Results and Discussion

The observation of the single tracks as they were being printed (i.e., melt tracks, Figure 4) revealed that none of these were an unsegmented semi-cylinder (perfect semi-cylinder). In series B, a segmented cylinder was observed to be stable until the slope angle β reached 50°. When this value was exceeded, the cylinder collapsed, which led to the formation of separate bulbs (balling phenomenon). In series A, however, none of the melt tracks created segmented or unsegmented cylinders; thus, the melt track was not even stable at β = 0°. The higher the β angle, the more segmented the track morphology became (Figure 4). In both series, the effect of melt-track disruption became more pronounced as the β angles increased. The melt tracks were cut and evaluated, and the results are presented in Figure 5.

In Figure 5a,b, values for the measured β angles (β_m_) are plotted against the β values. In series A, the β_m_ (β_measured_) matched the β values. In series B, however, the β_m_ values were lower than expected. These findings may indicate that the higher L_p_ caused the previous layer to partially re-melt, a process that was not compensated by an increase in laser speed L_p_. It was not possible to identify any STCSs for series B at β = 70° and for series A at β = 60°.

Increasing the L_p_ in series B allowed the melt-track width “L” to be maintained at a nearly constant value, while the L value was observed to decrease as β increased in series A. The L values in series B, however, were slightly higher, and their values were much less variable. The values of the melt-pool height (Figure 6c,d) showed that the melt track height increased as the β_m_ angle increased. As an increased contact width can support the stability of a melt pool, this increased height could negatively influence the stability of these samples. 

Even though the linear energy E_l_ was equal in both series, series B showed a larger area of STCS with respect to series A (Figure 7a,b). This was accompanied by larger radii, as presented in Figure 7c,d. These findings indicate that the increase in L_p_ melted more powder, which was supported by the increased area of the melt-track cross-section. Additional powder could originate from the previous powder layer. Increasing R and decreasing L could potentially reduce the stability of the melt pool and eventually lead to its destabilization.

The last measured parameters were the contact angles Θ_adv_ and Θ_rec_. In the case of perfect wetting conditions, the contact angles of the melt pool (and the melt track) showed very high values (>>90°). The lower these values, the lower the stability of the melt pool, and the more likely balling was to occur. Θ_adv_ decreased as a function of the β_m_ angle at a similar rate in both tested series; however, the overall values were higher in series B (Figure 8a,b). On the other hand, the Θ_rec_ in both series slightly increased with β_m_ (Figure 8c,d). The standard error was observed to be high, mainly due to the presence of non-melted particles attached to the melt track. 

An observed continuous decline in Θ_adv_ values (Figure 8a,b) suggests the presence of forces that pull the melt pool down the inclined surface. On the other hand, the low magnitude of changes observed in Θ_rec_ (Figure 8) suggests the presence of forces that help keep the cylinder stable along the incline. These findings may be attributed to the high surface tension of the molten steel [21]. The equation to determine the force f_p_ present at the given inclination is
(1)$$f_{p}=\rho V\sigma ({\cos\Theta }_{\mathrm{rec}}{-\cos\Theta }_{\mathrm{adv}})$$
where the surface tension σ = 1.75 N·m^−1^ [21] and the density ρ = 6.92 g·cm^−3^ at 1900 °C [22]. The volume of a melt pool V = A·λ, where λ = length of a melt pool.

Equation (1) was adopted in order to reflect the changing inclination angle such that both angles Θ_rec_ and Θ_adv_ would be linear approximations derived from the data presented in Figure 8.

As presented in Figure 9, the forces pushing the melt pool down the slope increased continuously and were lower when the parameters for series B were used. This was due to higher values of Θ_adv_ and L that were found for series B. Slippage, however, was not observed during the experiment, even though the magnitude of forces is comparable to the surface tension (7.85 × 10^−4^ N for λ = 400 μm; σ = 1.75 N·m^−1^ at 1850 °C [21]). In addition, the melt pool in the investigated set was not free to slide down the slope, as any contact between the melt pool and solid-state steel would cause an energy outflow and, eventually, the immediate solidification of the material. An analysis of the influence of surface tension requires the adoption of the approach developed by Yadroitsev, Bertrand, and Smurov [23]. These researchers performed single-track experiments using PBF-L/M and showed that the moving laser spot created an elongated pool of molten metal with a quasi-semi-cylindrical cross-section due to the strong influence of the surface tension (Figure 10). In such a set, the surface tension plays a great role in the melt-pool stability, particularly in a system with a size that typically does not exceed 500 μm. The unsegmented cylinder (Figure 10a) easily becomes unstable. Its segmented counterpart (e.g., meandering shape, Figure 4b, β = 10°) remains stable under a wider range of conditions if the disturbance increases the area of surface in contact with the base material [7] (Figure 10b). 

The model developed by Gusarov and Smurov [7,24] can be used to approximate the moving pool of molten metal to a liquid cylinder. It also subordinates its stability in proportion to the relation of the diameter to the length and to the specific angle Φ (Figure 10). According to the results of their analysis, a liquid semi-cylinder remains stable only when the following necessary and sufficient condition is fulfilled:(2)$$\frac{\pi D}{\lambda }>\sqrt{\frac{\Phi \left(1+cos2\Phi \right)-sin2\Phi }{2\Phi \left(2+cos2\Phi \right)-3sin2\Phi }}$$

In the limit of a circular cylinder, Φ=π gives
(3)$$\frac{\pi D}{\lambda }>\sqrt{\frac{2}{3}}$$
which is applicable at Φ>π/2.

For further calculations, the values derived from the linear approximations presented in Figure 5, Figure 6, Figure 7 and Figure 8 were applied. 

The plot of a stability factor Φ/π [7] against the β angle for melt tracks from series A and series B cylinders is presented in Figure 11a. The lower the value of this factor, the more stable the liquid cylinder becomes; this is supported by the observation that the melt tracks created in series B were more stable than those in series A. The maximum length of a stable liquid cylinder was calculated by inverting Equation (2), (Figure 11b). The higher the inclination angle of the printed surface (the higher the β angle and lower the α), the shorter the maximum length of the cylinder (Figure 11b). Even a small inclination causes a severe reduction in the maximum allowed length. This is because the radius grew while the cylinder width “L” remained constant (series B) or decreased (series A), causing the Φ value to drop, consequently destabilizing the liquid cylinder. Furthermore, it increased slightly (β > 50°), mainly due to the constant increase in the STCS radius. By comparing these data to the SEM observation in Figure 4, it is possible to identify the length of a melt pool at the steepest inclination angle at which it was still stable (10° for series A and 50° for series B). The values for the maximum stable length of a melt pool for series A and series B were 365 and 380 μm, respectively.

Once all these parameters are known, it is possible to plot the stability map for the investigated melt tracks for selected length “λ” values (Figure 12). Such a map incorporates two stability parameters, as shown by Gusarov and Smurov [7]: i) Φ/π, which reflects the relation between size and base width of the STCS, and ii) πD/λ, which reflects the relation between size and length λ of the STCS. Due to the technical limitations of the experiment, the length of the melt pool was not measured directly. Therefore, for the purposes of comparison, the values for the last stable melt pool were applied (at a 10° inclination angle for series A and a 50° angle for series B). In fact, the actual length for both series may change according to alterations in the inclination angle. Therefore, the measurement points for a known melt-pool length are presented in Figure 12 as solid circles or squares, whereas the circular or square outlines represent points under the assumption that the melt-pool length did not change. 

The increased L_p_ and L_s_ (series B) resulted in the migration of data points into the region of stability, findings that agree with direct observations made of melt tracks (Figure 4). This migration was caused mainly by: i) The increased value of the melt-pool base length “L,” which influenced the Φ/π factor, and ii) the increased size of the STCS, which resulted in an increase in the πD/λ stability factor value. The influences of these effects on the downskin quality are presented in Figure 13, Figure 14 and Figure 15.

The results of the analysis of the downskin quality (Figure 13) show that the lower the α angle, the more irregular the downskin surface was (i.e., rougher patterns). The downskins in the series B samples remained relatively uniform until α reached a value of 60°; once this value was exceeded, the downskin surfaces appeared to display qualitatively higher amounts of irregularity as compared to series A samples. At α angles between 90 and 60°, the higher laser L_p_ in series B provided better contact with the previous layer by re-melting it and consequently increasing the “L” value with respect to the series A samples. Furthermore, when the inclination angle increased, more powder was available to be molten, as presented in Figure 3. This increased availability, combined with higher laser power, resulted in a rapid increase in the roughness of the downskins, as observed in Figure 13 and Figure 14.

Roughness measurements (Figure 15) showed that the roughness increased as the inclination angle of an unsupported surface decreased. When the inclination angle α remained within the range of 90°–70°, the roughness of series A samples was higher in both tested directions. When the inclination angle α exceeded 70°, the roughness increased rapidly in series B samples and exceeded the roughness of series A samples in the direction perpendicular to the X–Y plane. The differences, however, were minor. As soon as α reached 60°, the roughness of series B increased rapidly in a direction parallel to the X–Y plane and exceeded the roughness of the series A samples. Much smaller differences were observed in the direction parallel to the X–Y plane. These findings indicate that, if the melt pool is destabilized and balling occurs, this is passed on to subsequent layers. Furthermore, this occurs naturally, as the subsequent layer can solidify only once the previous one is formed. When the inclination angle was lower than 60°, the roughness of the series B samples exceeded that of the series A samples. These results show that higher laser power results in high roughness as soon as the melt pool enters an area of instability (Figure 12). 

## 4. Conclusions

The influence of the melt-pool stability on downskin roughness was investigated in the PBF-L/M process using AISI 316L powder. The results show that

the area of the melt-pool cross-section increased with the inclination angle due to the availability of extra powder. A constant increase in the radius of the melt track coupled with a constant value of the melt-track width “L” contributed to the melt-pool instability, segmentation, and, finally, balling.surface tension forces are the main cause of melt-pool balling rather than gravitational pull; the influence of the latter is limited due to the limited slippage of molten metal.the stability of a melt pool can be increased by improving the connection between the melt pool and a previously formed layer and by increasing its radius as much as possible. Shortening the melt-pool length can increase its stability on an incline. The stability of the melt pool should be considered when unsupported, inclined surfaces are being printed. In industrial applications, these results would suggest that the printing parameters for the outline have to be modified along the decreasing inclination angle, i.e., the laser power has to be continuously decreased, while the spot speed should be continuously increased at the same time.The use of higher amounts of laser power may result in lowered downskin roughness, but only if the melt track is stable. On the other hand, the instability of the melt track causes a significant increase in roughness with respect to the counterpart factor of lower laser power. The results of this study show that the laser power should be adjusted and consequently lowered as the printing angles become increasingly steeper, mainly to avoid melting the powder that has been placed beneath the overhang.The factor of melt-pool instability was mainly found to influence the downskin roughness in the direction parallel to the X–Y printing plane, as the perpendicular direction was subjected to re-melting effects as the subsequent layers were built.Linear energy is not a factor that is suitable for comparing varying laser parameters.

## Figures and Tables

**Figure 1 materials-13-00808-f001:**
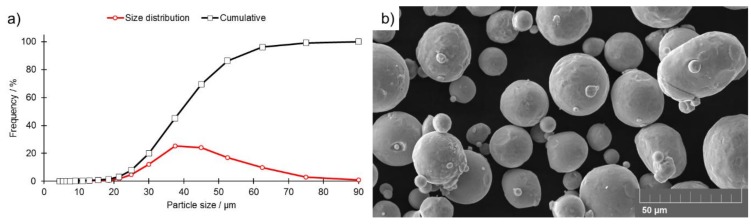
AISI 316L austenitic stainless-steel powder: (**a**) Grain size distribution, (**b**) SEM micrograph.

**Figure 2 materials-13-00808-f002:**
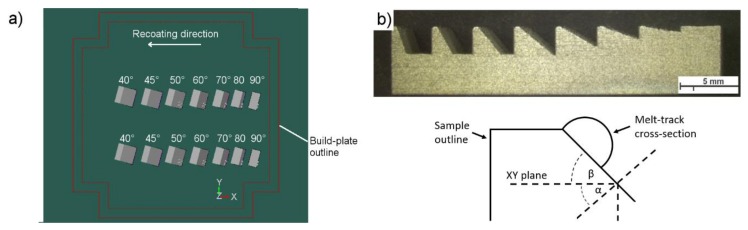
(**a**) An overview of the set of cuboidal samples and their location on the build-plate; (**b**) scheme of a sample used to simulate the increasing inclination angle, where α is the leaning angle of an unsupported surface and β is the inclination angle of a slope, with β = 90 − α.

**Figure 3 materials-13-00808-f003:**
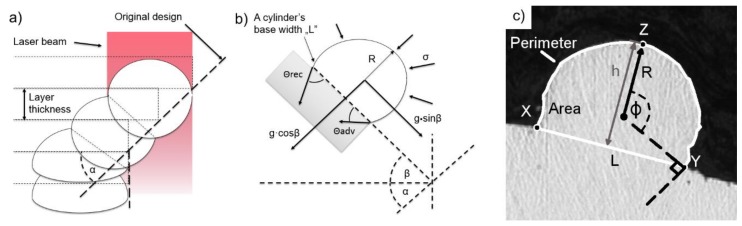
Schematics of the melt pool cross-sections: (**a**) Arrangement along the edge of the printed area; (**b**) forces acting on the melt pool when placed on an incline, and (**c**) display of specific points and parameters. L_t_—Layer thickness; g—gravitational constant; σ—surface tension.

**Figure 4 materials-13-00808-f004:**
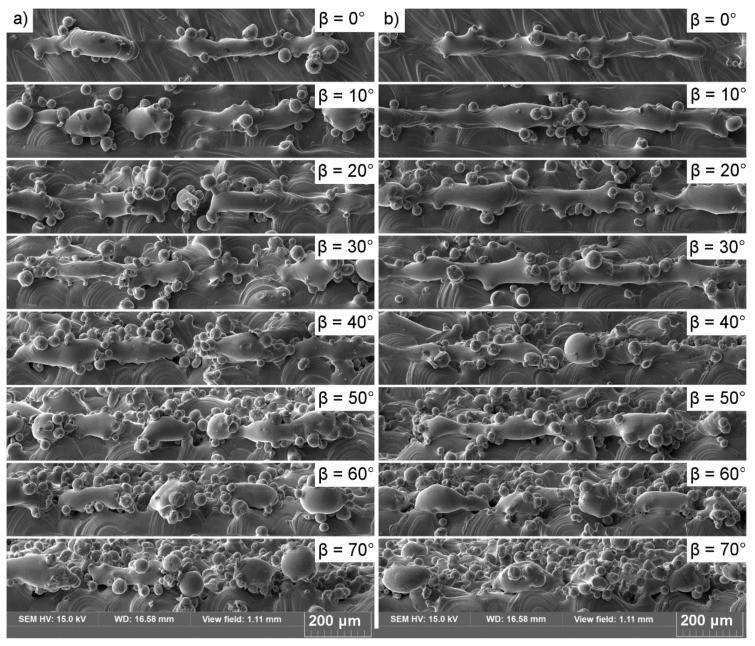
Top view of single melt tracks printed on slopes. (**a**) Series A; (**b**) series B.

**Figure 5 materials-13-00808-f005:**
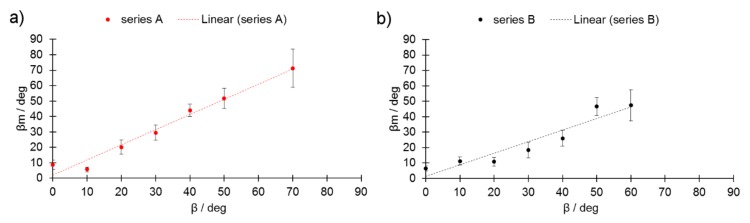
Measured inclination angle (β_m_) as a function of the designed one (β) for (**a**) series A and (**b**) series B.

**Figure 6 materials-13-00808-f006:**
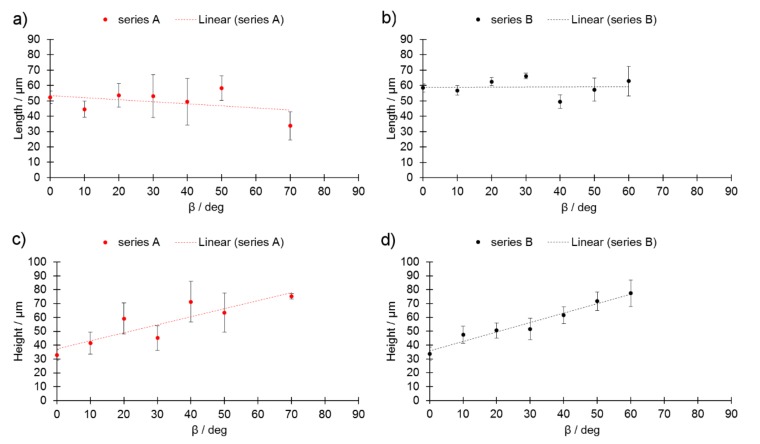
Melt-pool length (L) for (**a**) series A, (**b**) series B, and melt-pool height (h) as a function of β_m_ (**c**) for series A and (**d**) series B.

**Figure 7 materials-13-00808-f007:**
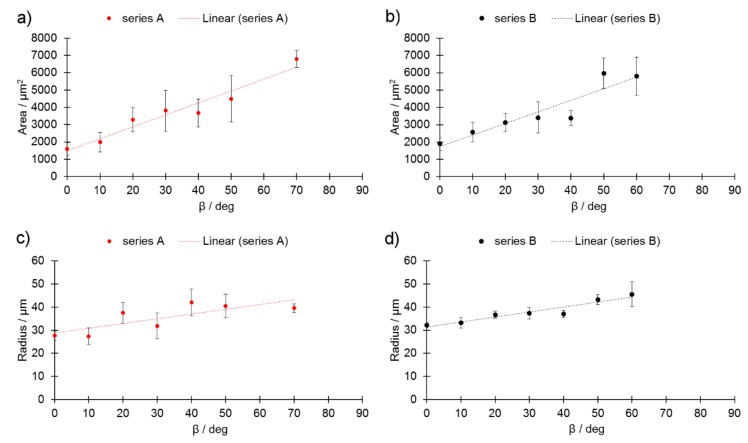
Area (A) for (**a**) series A and (**b**) series B, and radius (R) as a function of β_m_ (**c**) for series A and (**d**) series B.

**Figure 8 materials-13-00808-f008:**
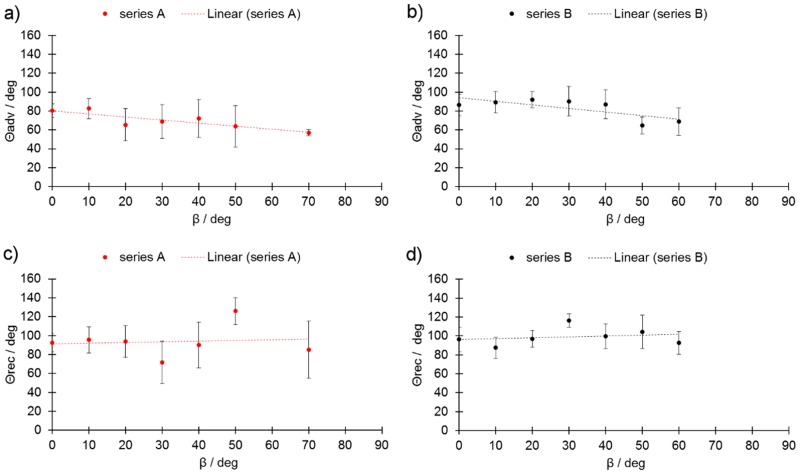
Advancing wetting angle (Θ_adv_) for (**a**) series A and (**b**) series B, and receding wetting angle Θ_rec_ (R) as a function of β_m_ (**c**) for series A and (**d**) series B.

**Figure 9 materials-13-00808-f009:**
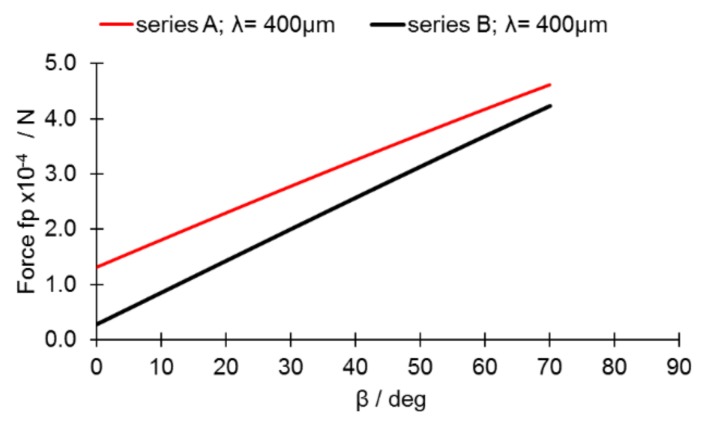
Forces (Equation (1)) acting on the melt pool for series A and B.

**Figure 10 materials-13-00808-f010:**
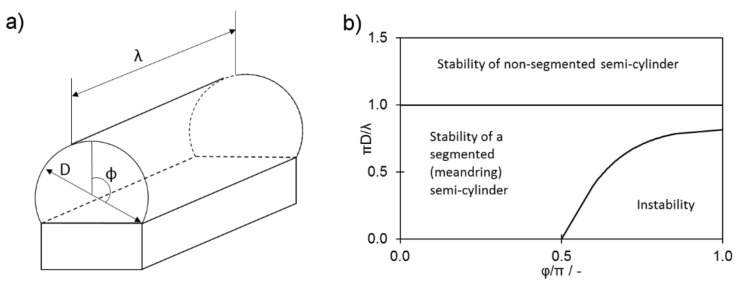
(**a**) Capillary stability of an unsegmented cylinder of a liquid on a solid substrate; (**b**) stability map based on [7]. λ is the length of a melt pool and D is the semi-cylinder (melt-pool) diameter.

**Figure 11 materials-13-00808-f011:**
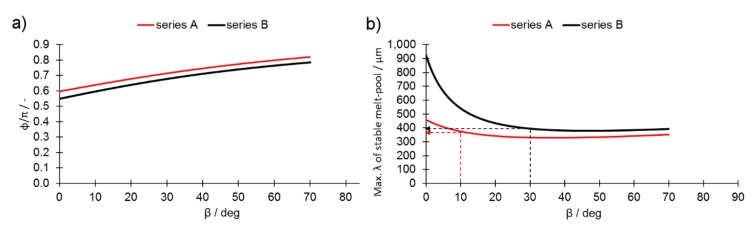
(**a**) Stability factor Φ/π as a function of β inclination angle; (**b**) maximum length of a stable liquid cylinder.

**Figure 12 materials-13-00808-f012:**
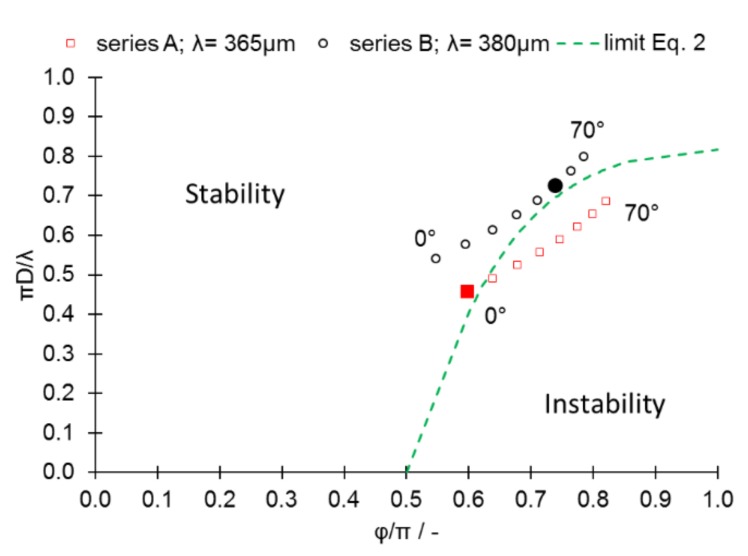
Stability map for series A and series B. The circular and square outlines represent values that were calculated, assuming melt-pool lengths of 365 μm for series A and 380 μm for series B. Solid circles or squares indicate measurement points for a known melt-pool length.

**Figure 13 materials-13-00808-f013:**
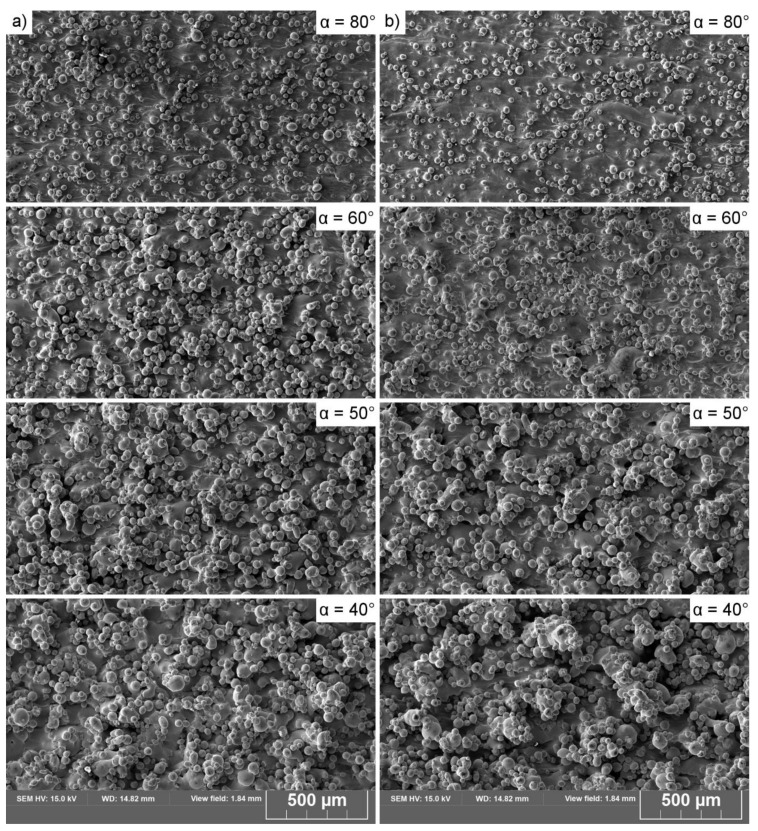
Representative overview of unsupported, inclined surfaces (downskins) from selected samples: (**a**) Series A; (**b**) series B.

**Figure 14 materials-13-00808-f014:**
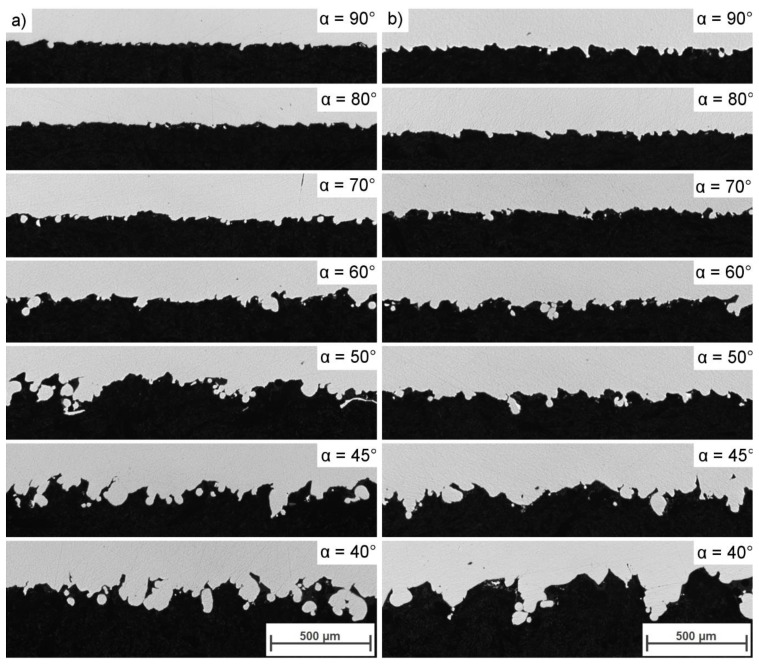
Cross-section of downskin areas: (**a**) Series A; (**b**) series B.

**Figure 15 materials-13-00808-f015:**
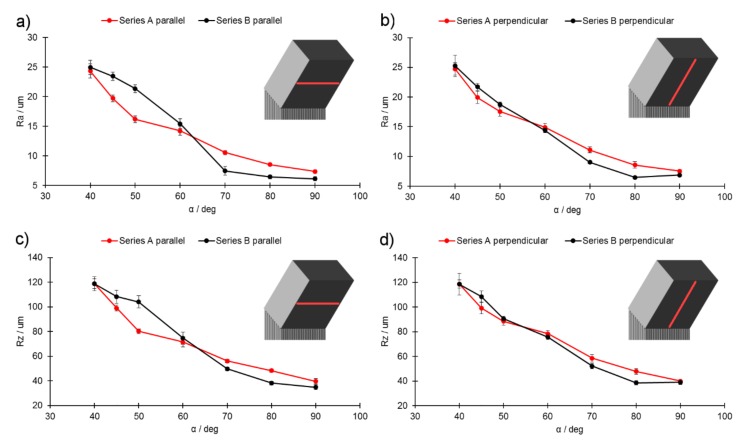
Comparison of roughness measurements for downskins: (**a**) R_a_, parallel to the X–Y printing plane; (**b**) R_a_, perpendicular to the X–Y printing plane; (**c**) R_z_, parallel to the X–Y printing plane; (**d**) R_z_, perpendicular to the X–Y printing plane.

**Table 1 materials-13-00808-t001:** Chemical composition of 316L stainless steel.

Element	C	Cr	Cu	Fe	Mn	Mo	N	Ni	O	P	S	Si
wt. %	0.03	17.5–18.0	<0.5	Bal.	<2.0	2.25–2.50	<0.1	12.5–13.0	<0.1	<0.025	<0.01	<0.75

**Table 2 materials-13-00808-t002:** Printing parameters for cuboidal samples.

Area of the Cross-Section	Beam Function	Remarks	Series A		Series B	
			L_p_	L_s_	L_p_	L_s_
Infill Parameters	Hatching	90 µm of hatching distance. Layer thickness = 20 μm.	160	960	160	960
Contour Parameters	Standard	n/a	100	770	100	770
	Up Skin	n/a	160	790	160	790
	Down Skin	n/a	70	990	85	1200
	Edge	Beam offset = 0	50	200	50	200

L_p_: Laser power/W; L_s_: Laser spot speed/mm·s^−1^; *under printing procedure of cubic samples, hatching was rotated by 33° for each consequent layer with respect to the previous one.

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
