# Peer review of "Stability of a Melt Pool during 3D-Printing of an Unsupported Steel Component and Its Influence on Roughness"

_materials, 2020, doi:10.3390/ma13030808_

Round 1

Reviewer 1 Report

The scientific work is well done. The approach is very interesting and gives some insight into the instabilities in down skin surfaces in LPBF.

The following things should be addressed when revising the article:

Figure 1: Aspect ratio seems off

Analysis equipment is stated in detail, please also state the used LPBF-equipment and if possible also the laser configuration (especially the focus diameter).

Please consider to include the chemical composition of the powder.

Please use same process abbreviations in the whole document. LPBF is good, PBF-L/M according to ISO ASTM 52900 may be even more appropriate. SLM should not be used.

If possible please mark the parameters h and L a bit more pecise in Figure 3.

Please describe how the powder was spread in the process for the single tracks and how laser positioning for the single laser tracks was realized.

The position of the part or the downskin surface in relation to the laser position may be a significant influencing factor. Please provide position and orientation of the specimen on the build plate. For future work a check on this influence may be appropriate. The difference between the values of beta may also be partly influenced by this effect.

Some elaboration or drawing may be good in order to understand the orientation of the roughness-measurement in Figure 14. Please also consider to use more detailed analysis methods for roughness in future papers. Figure 13 Alpha 45 and Alpha 40 seem to show a different type of systematic roughness. This could be addressed in future work.

Author Response

Dear reviewer,

Thanks for your time, work and interest. Enclosed in the attachment the document with the desired improvements.

kind regards

Benjamin

Reviewer 2 Report

The influence of inclination angle on stability, i.e. on surface rougness of LBPF manufactured samples was investigated by the authors.

Dear authors, please take into consideration the following suggestions:

Line 71 – „Gas-atomised AISI 316L austenitic stainless steel powder of was selected in this study (Figure 1).“ – to delete „of“

Line 73 – comma after 90o

„The series of samples was printed…“; „The series of single lines were printed…“; „…in the function of…; „…in function of…“; „The plot of … were presented…“; “The influence of … were presented…“; „…the main cause of … are the surface…“ – English language checking

mm/s or mm.s-1; please check other units of measurement

Figure 2 before Table 1? - it is mentioned before Table 1

Table 1 – too big with empties in the last row?

SLM – it is well known abbreviation, but it should be explained during the first appearance.

Please, compare the Lines 73 and 91, 92 – if the expression alpha plus beta = 90 degrees, then…

Line 89 – without bracket “…mm/s)”

Figure 2 - β = 90 – α instead of β = 180-α.

Line 130 – „βm values“ – “values” – not subscript?

Figure 6 – in the function of “β” instead of “βm”?

Line 170 – the number and unit of measurement are stick together (there is no space) – please check the other similar cases

There are many terms – unsegmented, nonsegmented, un-segmented – please use one uniformly (or molten cylinder, liquid cylinder).

Figure 9 and sentence in Lines 176 and 177 are contrary? (smaller, larger??).

Line 188 – please insert „Figure“ before 4b

Lines 210, 211 – the sentence is unclear

Lines 221 - 223 – the sentence is unclear (maybe to use comma, or two sentences?)

„Figure 12. Stability…“ instead of „Figure 12. stability…“

What is lambda?

Ra, Rz (sometimes using the subscript, sometimes not) – please, check other similar cases, Ls, Lp…

Line 266 “4. Conclusions” instead of “Conclusions.”?

Very important: to add to Conclusion – applicability of the investigation results in industry.

Conclusion – using bulleting or numbering can highlight the conclusions.

Please check the numbering of references, there are 25 cited references (not 24).

Yaroditsev or Yadroitsev?

DOI number – please check uniformity in writing.

Author Response

Dear reviewer,

Thanks for your time, work and interest. Enclosed the document with the desired improvements.

kind regards

Benjamin
